# Mapping of Deep Neural Network Accelerators on Wireless Multistage Interconnection NoCs

**Yassine Aydi \***, **Sirine Mnejja**, **Faraqid Q. Mohammed** and **Mohamed Abid**

CES Laboratory, National Engineering School of Sfax, Sfax 3038, Tunisia; sirine.mneja@enis.tn (S.M.);
faraqid@bauc14.edu.iq (F.Q.M.); mohamed.abid@enis.tn (M.A.)
* Correspondence: yassine.aydi@enis.tn

**Abstract:** In the last few decades, the concept of Wireless Network-on-chip (WiNoC) has emerged as a promising alternative for Multiprocessor Systems on Chip (MPSOC) to achieve reliable and scalable communication. Worth recalling in this regard is that our research team has already designed, verified and evaluated Multistage Interconnection Networks (MIN) in this field. With respect to the present work, we consider proceeding with further exploring our thoughts on this research area. Firstly, we propose the design and performance evaluation of a hybrid (wireless/wired) MIN, analysing how this augmented network can potentially improve not only the average delay, but also energy consumption. Secondly, we continue with examining the implementation of our advanced DELTA-based MIN architecture on Deep Neural Network (DNN) accelerators, while accounting for its potential regularity and scalability in simultaneously maintaining an effective power efficiency and lower latency throughout the DNN operating process. In this context, several metrics have been evaluated in regard to three DNN application cases through implementation of their main respective modules.

**Keywords:** Network-on-chip; wireless/wired NoC; Multistage Interconnection Network; DNN accelerators





## 1. Introduction

In recent times, there has been a significant interest from both industry and academia in developing efficient Network-on-chip (NoC) designs. These NoCs facilitate interconnectivity among fine-grained computing cores (CCs), which operate in parallel to enhance the overall computation process [1,2]. It is estimated that the number of integrated PEs in a System on Chip (SoC) would exceed a thousand by the next decade [3]. In this regard, architects have proposed innovative NoC designs that enable efficient and dependable communication for massive parallel applications. Equipped with specific built-in inter-core data routing, these designs are intended to help in noticeably reducing packet traffic and transmission delay. Following these advancements, other interconnect technologies have emerged, including Wireless/RF, Hybrid Wireless, and Photonic NOCs [4–7]. These NoCs provide a robust platform for scaling the number of cores in a chip [8].

Multistage Interconnection Networks (MINs) have been utilized for communication in supercomputers [9], the MasPar [10], and CRAY Y-MP series [11,12]. Additionally, MINs are used in NoCs to connect computing cores (CC) to memory modules or CC to CC in parallel architecture. They are also applied to manage IO data exchange, as a crucial aspect of massively parallel SoCs. Among the special features characterizing a MIN are its topology, switching strategy, routing algorithm, scheduling mechanism, fault tolerance [13,14], and dynamic reconfigurability [15]. Other keyfeatures include the constant number of hops separating any pair of terminal nodes, regularity, and scalability when increasing the number of cores in a chip.

In the domain of massive parallel applications, Deep Neural Networks (DNNs) have been utilized in various fields such as pattern recognition, prediction, and computer vision [16–18]. These accelerators have traditionally been deployed via ASIC or FPGA designs. However, the performance of artificial neural networks is often constrained by significant communication overheads and storage requirements. For the purpose of improving the reducing interconnection complexity, a regular and scalable NoC stands as a convenient applicable mechanism fit for implementation [19–21]. Owing to their noticeable data computation and communication distinguishing capacities, DNN accelerators are liable to provide rather effective computational flexibility, design simplicity, and high scalability advantages. Most of the recently elaborated research works have been predominantly focused on exploring efficient NoC topologies likely to minimize power consumption and maximize bandwidth performance through the incorporation of DNN computing. By integrating accurate core mapping and robust routing algorithms, these NoC designs turn out to be capable of maintaining highly flexible communication among PEs, enabling the handling and processing of diverse DNN models with varying data flows by means of a single architecture. Moreover, the system's computational power and performance features could be adjusted to match and cope with the underlying DNN models. Hence, the present work is intended to enrich the relevant literature by providing a twofold contribution. On the one hand, it puts forward a novel Delta-MIN-based design of a hybrid (wired/wireless) NoC. On the other hand, it serves to maintain an effective performance evaluation of DNN accelerators by integrating them within the Delta MIN framework.

The remainder of this paper is organized as follows: Section 2 provides a general overview of Multistage Interconnection Networks (MINs), along with their respective properties and implementation process. Section 3 explores the review of the related literature in this cross-topic area. Section 4 presents the design of the advanced Delta MIN-based hybrid Network-on-chip. Section 5 examines the implementation procedure of the NOC developed on the Deep Neural Network accelerator. As to the final section, it is devoted to depicting the major conclusions that are drawn.

## 2. Taxonomy of Mutistage Interconnection Networks (MINs)

This section provides highlights of the networks applied in the design process of a Hybrid Delta MIN for a multicore system on a chip.

### 2.1. MIN Graph Cartography

The MINs applied in our architecture involve N input and N output nodes, and they are mapped using $r \times r$ switches. These dynamic architectures have $N/r$ switches at each stage level, with $\log_r(N)$ stages of switches represented by $d$. The different stages are interconnecting on permutation functions. As illustrated on Figure 1, a cross-bar chart of an abstract model of an $N \times N$-size MIN is depicted, with $r$ being equal to 2. In a MIN, a pathway between a source and a destination is easily recognised by straightly enabling the stage $i$ corresponding switch once the $i$th bit of the destination address turns out to be equal to one; otherwise, it remains in a cross-mode state.

### 2.2. Banyan Property

Figure 2 illustrates a taxonomy of MINs , as envisioned throughout the present section. A banyan MIN is a type of design that is intended to guarantee the preservation of a single path between each source and final target. Typically, a banyan MIN with a size of $N \times N$ is constructed using $r^2$ crossbars. According to the banyan MIN set specification, an interesting class recognised as Delta networks can be mapped [22].

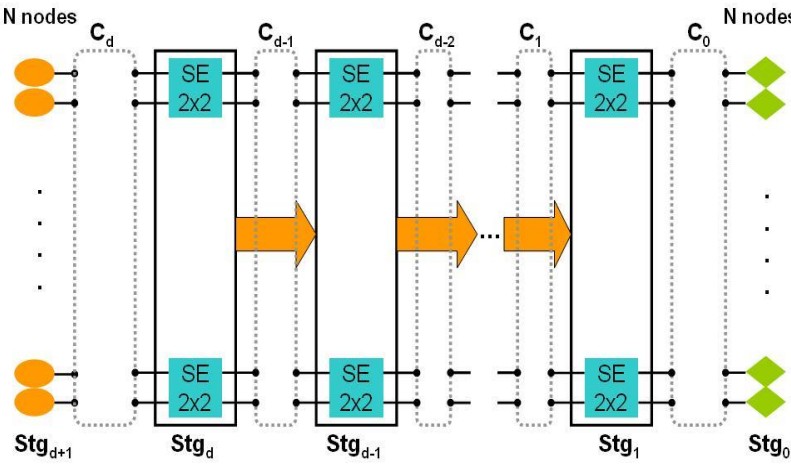

**Figure 1.** A detailed model of Multistage Interconnection Network (MIN).

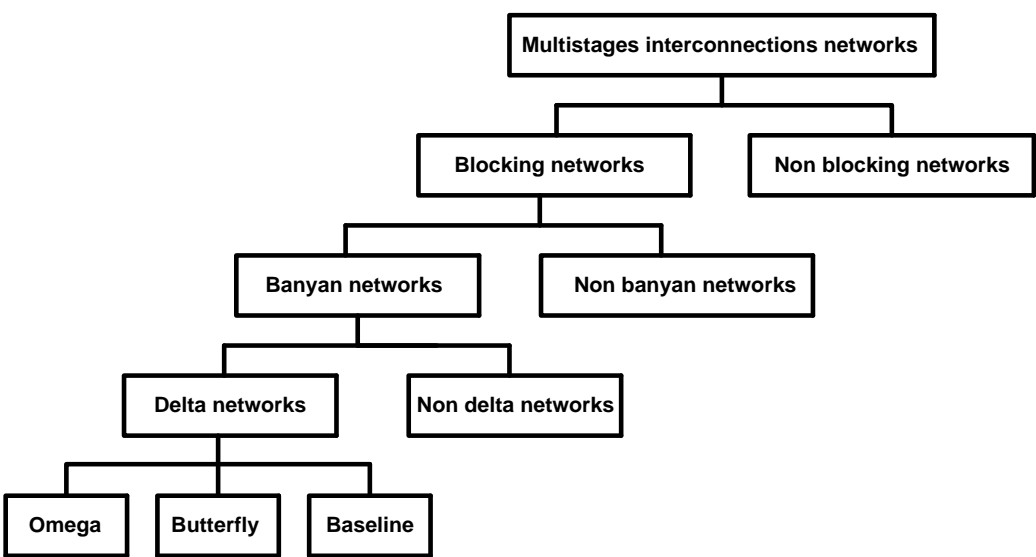

**Figure 2.** Taxonomy of MINs.

### 2.3. Delta Class MIN

Various MINs display noticeable differences as to the switch stages'connecting wires. A comparative analysis of a diversity of Delta MINs (Figure 3) is available in ref. [23]. With respect to our advanced architecture, $i$ denotes the $i$th the output of a crossbar in a MIN, while $C_j$ designates a stage $j$ relating crossbar. Accordingly, we define the Delta property in the following way: if one input of $C_j$ is linked to the output of $C_{j-1}$.

A comparative study of various Delta MINs, including Baseline, Omega, and Butterfly, is provided in Table 1. The most common modes of link permutations, performed on a $2 \times 2$ switch-bearing MIN with elements, include the perfect (represented as $\sigma$), butterfly (represented as $\beta$), baseline (represented as $\delta$), and identity (represented as $I$) permutation, as detailed below:

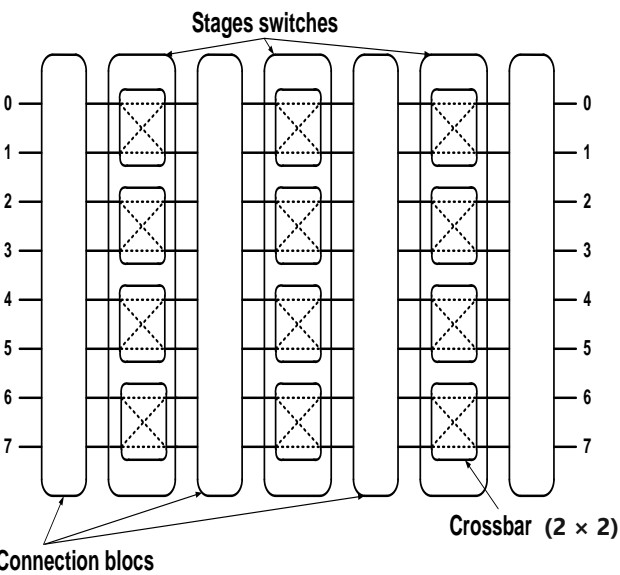

**Figure 3.** A Delta MIN model with eight nodes.

- The perfect shuffle: it is a kind of a bit-shuffling procedure, wherein, the $k$th bit of an input sequence $(x_{n-1}x_{n-2}\ldots x_1 x_0)$ is swapped with the $(k-1): \sigma^k(x_{n-1}x_{n-2}\ldots x_1 x_0) = x_{n-2}\ldots x_1 x_0 x_{n-1}$.
- The butterfly: permits the $k$th bit of the input sequence $(x_{n-1}x_{i+1}x_i x_{i-1}\ldots x_1 x_0)$ with the $(k-1)$th bit, while preserving the order of the other bits: $\beta_i^k(x_{n-1}x_{i+1}x_i x_{i-1}\ldots x_1 x_0) = x_{n-1}\ldots x_{i+1}x_0 x_{i-1}\ldots x_1 x_i$.
- The baseline: swaps the $k$th bit of the input sequence ($x_{n-1}x_{i+1}x_i x_{i-1}\ldots x_1 x_0$) with the $(k-2)$th bit, preserving the order of the other bits: $\delta_i^k(x_{n-1}x_{i+1}x_i x_{i-1}\ldots x_1 x_0) = x_{n-1}\ldots x_{i+1}x_0 x_{i-1}\ldots x_1$.
- The identity: helps maintain the input sequence relevant mapping: $I(x_{n-1}x_{n-2}\ldots x_1 x_0) = x_{n-1}x_{n-2}\ldots x_1 x_0$.

A summary of the three Delta MINs associated swapping functions over stages, as previously designed, verified, and evaluated by our research team, is illustrated in Table 1.

**Table 1.** Swapping Delta MINs Functions.

| Links | Stage 0 | Stage $(d+1)$ | Stage $k \in [1\ldots d]$ |
|---|---|---|---|
| Baseline | I | I | $\delta_{(d-i)}^k$ |
| Omega | I | $\sigma^k$ | $\sigma^k$ |
| Butterfly | I | $\sigma^k$ | $\beta_{(d-i)}^k$ |

## 3. Related Works

A great deal of research has recently been conducted in the area of multicore systems on chips, particularly the relevant design and interconnection evaluating architecture, in a bid to construct optimally effective communication platforms. A major area of growing interest has been the design of novel alternatives to Network-on-chip (NoC) fit for implementation with massive computing architectures entailing reliable data exchange between cores and memories. NoCs are capable of bearing an increasing number of computing cores on a single chip, thereby achieving high levels of parallelism and speeding up execution time. Contrarily, however, conventional shared buses consume greater amounts of energy due to the complexity of wire interconnections on the chip, thus, bringing about only lower throughput levels. Additionally, the temporary registers implanted in shared buses usually consume a greater area and higher energy levels, resulting in poor scalability. This is likely to stand as a major hindrance to the maintenance of effective communication via

future Multi-Processor Systems on Chip (MPSoC) designs [24–28]. To cater to these needs, Multistage Interconnection Networks (MINs) have emerged as a potential solution for the increasing demand for scalability and reliability in static Network-on-chip (NoC) architectures, to meet the exponential growth in massive parallel computing. The evaluation of MINs can be performed based on various metrics, including energy, area, throughput, fault-tolerance, network complexity, and cost-effectiveness. Most often, the functional formalization of MIN-based networks has been developed by identifying intrinsic properties of all MIN topologies. It has also been validated via the ACL2 theorem, to prove that the environment does actually comply with the network formal specification perquisite of integrating architectural parameters, with a significant impact on design costs [29,30]. Comparative studies have also been established to assess MIN systems in relation to static topologies, mainly in terms of reliability, performance. A comprehensive review of MIN systems in matters of reliability, fault-tolerance, and cost perspective evaluative aspects is available in the relevant literature [31,32].

As to the MPSoCs with heterogeneous cores, however, they have been designed to implement irregular topologies with hybrid wired/wireless interconnection [33]. Other research studies have been focused on comparing MINs architectures with other topologies [34–36]. In ref. [36], V. Dinh et al. introduced extra features to the Noxim simulator [37] to evaluate particular applications in an NoC system. They proposed a specialized design for relevant basic routing algorithms to explore their impact on performance metrics. Their major aim was to substitute longer channels with single-hop Radio Frequency transmissions among routers.

With the significant increase in computational requirements necessary for operating and boosting the various related applications, researchers are now experimenting with the implementation of artificial intelligence techniques in System on Chip designs to further enhance their performance. A major suggested approach consists in integrating Deep Neural Network (DNN) accelerators in NOC-Based MPSOC to manage computation and communication within the chip, thereby, enhancing computational flexibility and scalability [38]. The effectiveness provided by this integration process rests on its accounting for various network-associated features such as routing and scheduling strategies, node partitioning, data packetization, buffer sizes, QoS, and others. To achieve an effective adequation between the reconfigurability features and performance constraints, the accelerators need to be coupled with a structural network mapping. Moreover, by deploying parallelized multicore, the speed of operations over the chip could be improved even more. In this respect, several studies have attempted to explore the deployment of Deep Neural Network (DNN) in order to achieve highly efficient on-chip interconnects. For this purpose, crossbar communication is commonly used owing to the regularity and scalability of DNN operations [39,40]. Mesh, tree, and Clos networks have also been implemented with DNN accelerators to improve chip performance through efficient memory access, low-latency multicast communication, power efficiency, and computing flexibility [41–43]. Regarding reconfigurable architectures, studies have suggested replacing the crossbar with wire switchers in experimentation. Different types of interconnections, such as optical, wireless, or 3D networks, have recently been proposed and applied in DNN computing [44,45]. Another mapping algorithm tested for its effectiveness has also been put forward in ref. [46], while a communication-on-chip system focused on topology reconfiguration has been advanced in ref. [47].

As already stated, one could well note that the Multistage Interconnection Networks (MINs), classified into regular, scalable, and reconfigurable topologies, do not seem to be treated in terms of chip communication improvement purposes through wireless nodes, and appear to be exclusively evaluated through the perspective of applying Deep Neural Networks accelerators.

In recent research, Network-on-chip can be used as a communication backbone for chip multiprocessors, to improve the computational performance of neural network accelerators by dissociating communication and computing data. NoC features namely, scalability,

efficiency, reliability, and modularity, that enable computing components to exchange a huge amount of data, ultimately achieving enhanced energy efficiency and performance.In this context, the integration of CNNs or DNNs on NoC platforms have been illustrated so far [45,48–54]. Several hybrid parameters have to be carefully considered in the mapping process. First, the clustering of layers on computing elements (PEs) is managed by mapping algorithms. On the other hand, the NoC is specified by its topology, which defines the placements of routers and links, routing algorithm, and memory scheduling. Consequently, according to this mapping process, computation is executed on PEs as they routed over the NoC following an adapted DNN dataflow. In ref. [55], Chen et al., propose a simulator, baptized NN-Noxim. This tool details several constraints, such as classification precision, energy, and transmission delay, based on a hybrid configuration related to the DNN mapping model and NoC parameters. Neuron computation and data communication in the fully connected layers have been simulated and evaluated. This simulator has been extended in ref. [56] to integrate convolution and pool layers. However, these developed tools only support static NoCs, such as the mesh topology.

In the following section, we present our envisioned novel design of the Delta MIN-based NoC that incorporates wired and/or wireless interconnection. We also provide a relevant performance evaluation process. The assessment procedure has actually proven the noticeable benefits brought about by the idea of integrating radio hub nodes into the process of ensuring resilient and adaptative communication among the Delta MINs different incorporated stages [48].

## 4. Design of Hybrid Network-on-Chip Based on Delta MINs

In this section, the design flow of our advanced hybrid (Wired and Wireless) communication architecture based on Delta Multistage Interconnection Network is thoroughly detailed. The model is conceived to involve a number of relevant parameters, particularly, a special topology and structure, traffic model, dynamic behavior, and simulation. A set of metrics has been configured and evaluated. The model is implemented on the Noxim simulator. An extended version of Noxim is used, which includes radical improvements, related mainly to signals mapping, wireless communication within Delta MIN stages, and a specific routing algorithm to maintain flexible data transfer between cores (Figure 4).

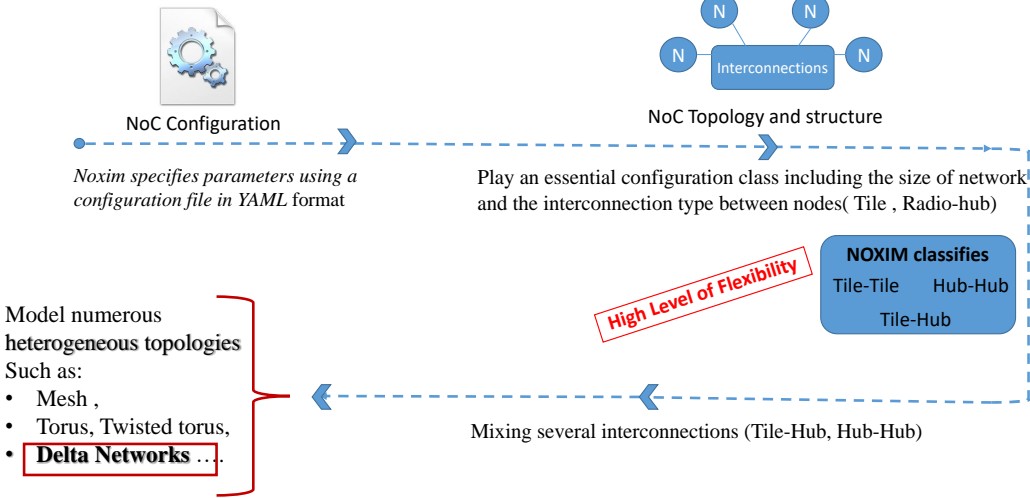

**Figure 4.** Noxim Simulator Design Flow.

The Delta MIN has been implemented by adding new signal mapping using SystemC to allow for various network topologies to be constructed (e.g., Butterfly, Omega, and Baseline) (Figure 5). Another required extension has been the design of the switch nodes, necessary to manage routing alternatives in the network. After introducing the topology, necessary for maintaining connection between the links and switches, we proceed with

incorporating radio nodes to enhance the wireless feature option throughout the Delta MIN networks. It is worth noting that wireless nodes represent switching components capable of exclusively maintaining a unique hop communication with distant switch nodes, which entails several hops to pass through a wired network configuration (Figure 6).

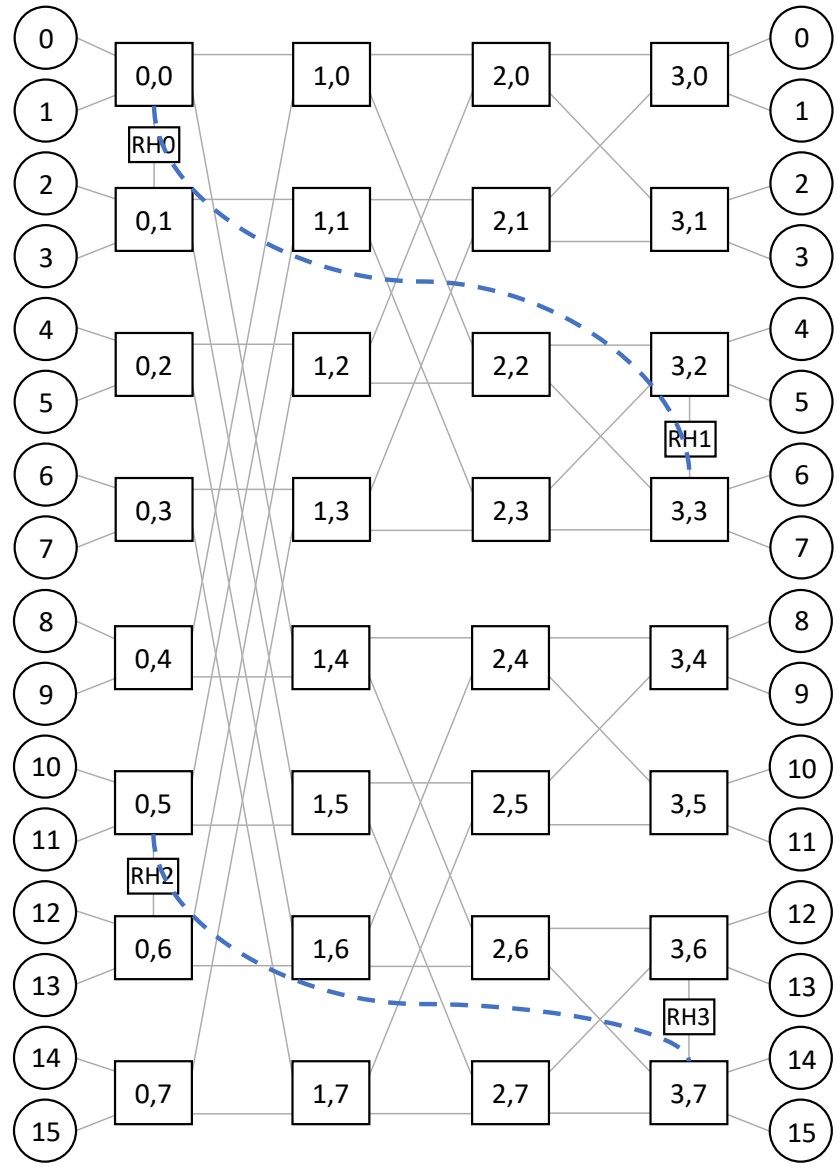

**Figure 5.** Wired and Wireless Communication on Delta MIN.

Then, we resume by introducing radio nodes, respectively, to the source and to the target. Subsequently, we set up transmission channels shared by these nodes, maintaining that they do not interfere with each other. To monitor the effect of increased numbers of wireless nodes on the architecture, a set of configuration models, Ti, has been implemented, each generating a single communication flow between the source and the target. Noteworthy, also, is that each data stream is routed via an ad-hoc communication channel, a critically useful alternative in our study to monitor wireless communication in Delta MINs networks. The successive steps, necessary to transmit data between switch and radio-hub are displayed in Figure 6. A radio node is connected with the switch through its wired gate. The information passing among radio nodes is accomplished via radio channels in which flow control is managed through a token implemented on the medium access control unit (MAC). A radio hub can route data using several communication channels. Connection is maintained by means of a token ring algorithm. Thus, a radio node can exchange informa-

tion through the channel's related token ring. Obviously, a radio node must capture the token to reactivate a transmission through a specific communication channel.

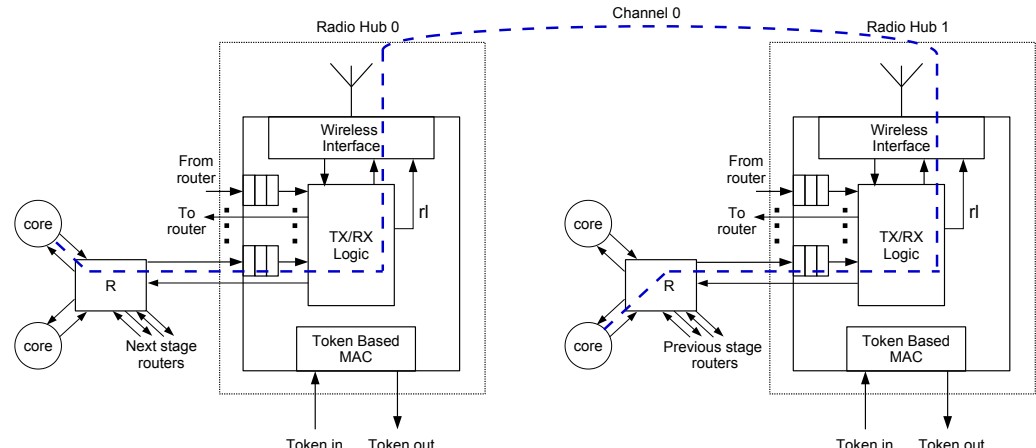

**Figure 6.** Communication on Delta MIN.Detail of wireless communication by means of two radio-hubs sharing a wireless channel connected to switches in the first and last stage.

After that, three traffic model patterns (Hot-spot, Fixed, and Random) are selected to assess the efficiency performance of the envisioned communication scheme. Each of these scenarios would be applied in accordance with various Network configurations (Table 2). Our main motivation behind opting for these parameters lies in ensuring the truthiness of values, as already integrated and explored in several previously published research works. The network size denotes the number of cores liable to generate traffic to activate the computation processing. For each Network configuration, a Mesh of switch nodes is needed to implement the MIN communication architecture.

**Table 2.** Simulation Parameter Space.

| Parameter | Value |
| --- | --- |
| Network size [cores/(switches × stages)] | $32/(16 \times 5), 64/(32 \times 6), 128/(64 \times 7)$ |
| 64/256 radio-hubs number | 4, 8, 16 |
| 1024 radio-hubs number | 16, 32, 64 |
| Switching technique | Wormhole |
| Radio Access Control Mechanism | Token Packet |
| Wireless data rate [Gbps] | 16 |
| Packet length [flit] | 8 |
| Flit size [bit] | 64 |
| Router input buffer size [flit] | 4 |
| Radio-hub input, antenna buffer size [flit] | 4 |
| Simulation Time | 100,000 cycles |
| Repetitions | 10 |

Figure 7 displays the average delay measurements reached on matching several network sizes and traffic patterns generated by following four wireless configuration profiles ($T1, T2, T3, T5$). As a first stage of the analysis, the effect of increasing the number of radio-hubs for a defined traffic configuration is evaluated. Accordingly, the random traffic scenario ($TT - 32/64/128$), for instance, turned out to highlight that increasing the number of radio-hubs (switching from $T1$ to $T5$) appeared to display a minor effect on average delay, particularly, at the breakpoint where saturation occurs.

Furthermore, it has been spotted that, for certain traffic models, each node randomly chooses a destination, the effect of the transition from profile $T1$ to $T5$ is only perceptible when the wireless option is activated, and the improvement in the saturation breakpoint

would be justified by a pressure alleviation within stage switches of the MIN communication platform. Finally, however, it is worth noting that a significant amount of data exchanged may skip the wired switches nodes when the number of radio-hubs increases.

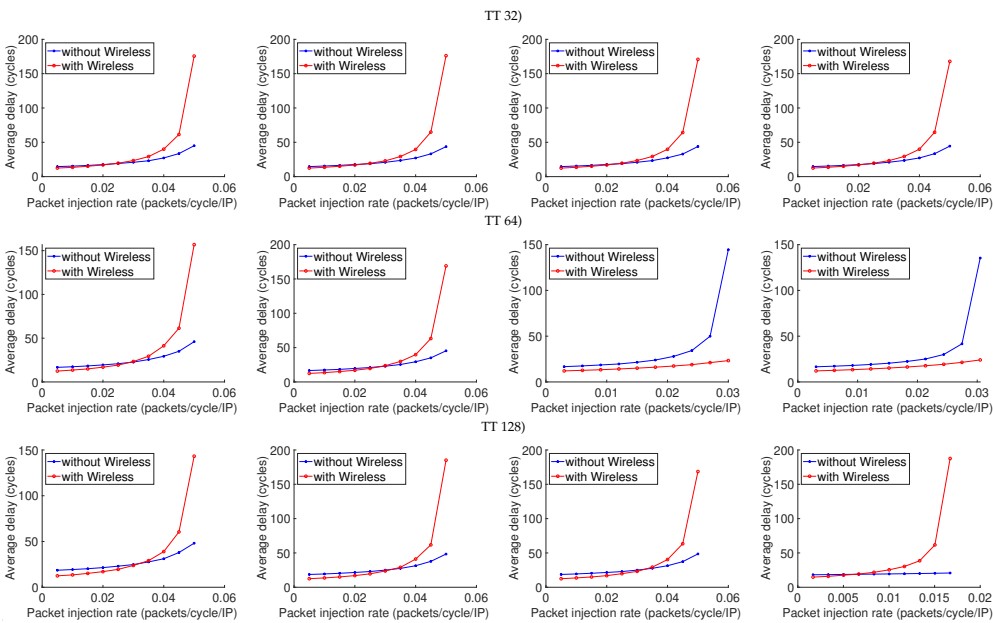

**Figure 7.** Performance Evaluation of Hybrid Delta MIN for table-based (TT) traffic for 32, 64, and 128 cores.

## 5. Evaluation of DNN Accelerators under Delta-Based MINs

Implementation of the DNN accelerators is fit for application, particularly with ASIC or FPGA designs. In the present section, we highlight the importance of introducing the Interconnection process into machine learning operations, in conformity with the exponential growth of parallel computation on chip. Figure 8 illustrates a sample example of Neural Network architecture dubbed *LeNet-5*. Designed to recognize handwritten digits, the scheme has been used by several banks to identify hand-written numbers on checks. Every number is digitized into $32 \times 32$ pixel grey scale input images. This neural network is managed on a seven-level convolutional network.

The subsequent section provides an extensive account of the evaluation process for the DNN accelerators within the framework of our hybrid Delta MINs. This involves incorporating the main modules into a NoC-based NN accelerator. To the best of our knowledge, we have undertaken a comprehensive investigation, focusing especially on the necessary steps required to conduct the experiments and perform an in-depth analysis. This analysis involves conducting a comparison between the Delta MIN design and a parallel Mesh topology.

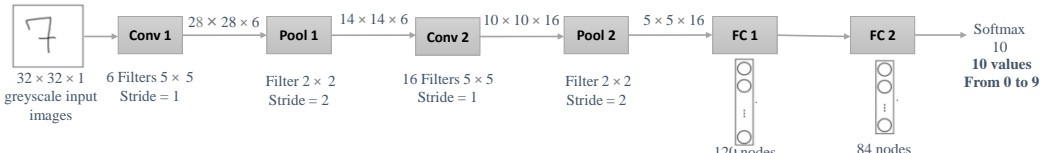

**Figure 8.** NeuralNetwork LeNet 5 Design.

Experimental analyses are performed on a number of the NoC architectural characteristics, including the topology and scale of the network, to assess the impact of Neural Network inference on latency end energy levels of the chip-based communication. In the context of interconnected architecture, a node can be a processing element (PE), or a memory (M). The PE plays a fundamental role in the performance of the computational

engine used in the NoC-based DNN accelerator. Indeed, it is heavily deployed at the level of the three main layers of the DNN, more specifically, the (conv., max/aver, and fully conn.) layers. The special mapping process of each layers will be specified in the NoC, along with the relevant identification of the data flows. We will also provide a detailed explanation of the flowing process of two main data paths: intra-NoC or data flows between the DNN and NoC-based processors, as well as the off-NoC or data flows maintained between the DNN–NoC and the main memory. At the level of the convolutional layer, a set of filters and the input feature map represent the inputs of the convolutional layer. Figure 9 outlines a convolution layer flow using six filters. In this example context, a 4 × 4 Mesh is applied as an NoC configuration involving a single shared Memory and fifteen PEs. At the first step level, the input feature map loading process is being drawn from the main memory. The MI is allotted the task of sending the input feature map to the PEs required at this layer's level. The filters are loaded from the main memory at the second step level and then sent to a particular PE. After getting all the necessary filters, each PE undertakes to compute an output feature map channel. In the ultimate step, every PE encloses all the filters to undertake the computing process of a fit channel out of the output feature map to be stored back in the main memory. The output feature map would stand as the input feature map for the next neural network-associated layer.

Both the average pool and max pool are considered for a pooling layer. We consider a $1:n$ mapping, binding the processing elements (PEs) and the feature map channels. It is worth specifying, at this level, that a PE is able to operate several feature map channels, i.e., every PE receives the input feature map channel that is currently stored in its local memory. Thus, no PE-to-PE traffic would persist. As already stated, the fetch of the feature map channel could be completed at the level of the main memory if the local storage element is not large enough. At the pooling layer level, the operations performed are either average or max types of operations among the input feature map lying data. The amount of operations depends on the input feature map size, the layer scale and stride parameters. For the fully connected layer level, the output neurons bear inputs corresponding to the size of the input feature map. In this context, we consider establishing a $1:n$ mapping between PEs and neurons, denoting that a PE can process multiple neurons. Thus, every PE is required to fetch a number of weights, from the main memory that corresponds to the input feature map size, as shown in Figure 10.

Designing an efficiently reliable and resilient interconnection for NoC architecture remains an important challenge, especially for applications with real-time constraints, high throughput, and power efficiency needs. To resolve these issues, various specific NoC technologies, particularly Photonic NoC, Hybrid, and Wireless, have been proposed, each displaying proper trade-offs in terms of throughput, communication latency, and programming complexity. A survey of NOC technologies developed to meet industrial requirements is presented in refs. [27,57]. With respect to our study context, our focus of interest lies mainly in investigating the area of Wireless NoC [58–60] (WiNoCs), initially designed to provide promising solutions to the challenges faced by traditional NoC architectures. In general, Wi-NoCs involve an enhanced model of radio-hub switches with a wireless interface, allowing for radio transmission to be maintained within the chip. While Wi-NoCs have been explored for common direct topologies, their implementation in Delta MIN-based topologies still remains an unexplored domain, to the best of our knowledge. A key advantage of Wi-NoCs is their low power consumption, which can be further improved by effectively managing power-off wireless routers when they are in an idle state, as highlighted in ref. [61]. Indeed, WiNoCs offer high bandwidth availability and reliability, and the reliability of wireless links can be improved by implementing an optimum-radiation phased array antenna, as demonstrated in ref. [62]. While hybrid wired/wireless links are more common in WiNoCs, a distinctive pure wireless link topology has been introduced in ref. [63]. The special feature distinguishing these solutions lies mainly in the graph of links substitution and/or the distinct architecture of the mapping of wireless nodes. The Experimental platform is a reconfigurable NoC-based machine learning, used to as-

sess different configurations of communication architectures in terms of performance and energy. A refined version of a high throughput accelerator for pervasive convolutional neural networks (CNNs) and Deep Neural Networks (DNNs), baptized DianNao [49], has been mapped into the computing module of Noxim [50] simulator. We focused on this accelerator, due to the wide range set of applications. Also, considering the implementation of large-scale layers composed of millions of synapses, the flexibility of interactions with memory, and by exploiting locality at the registers placed close to processor elements, our study broadly evaluates WiNoC-based DNN accelerators by investigating the design space that encompasses various reconfigurable parameters.

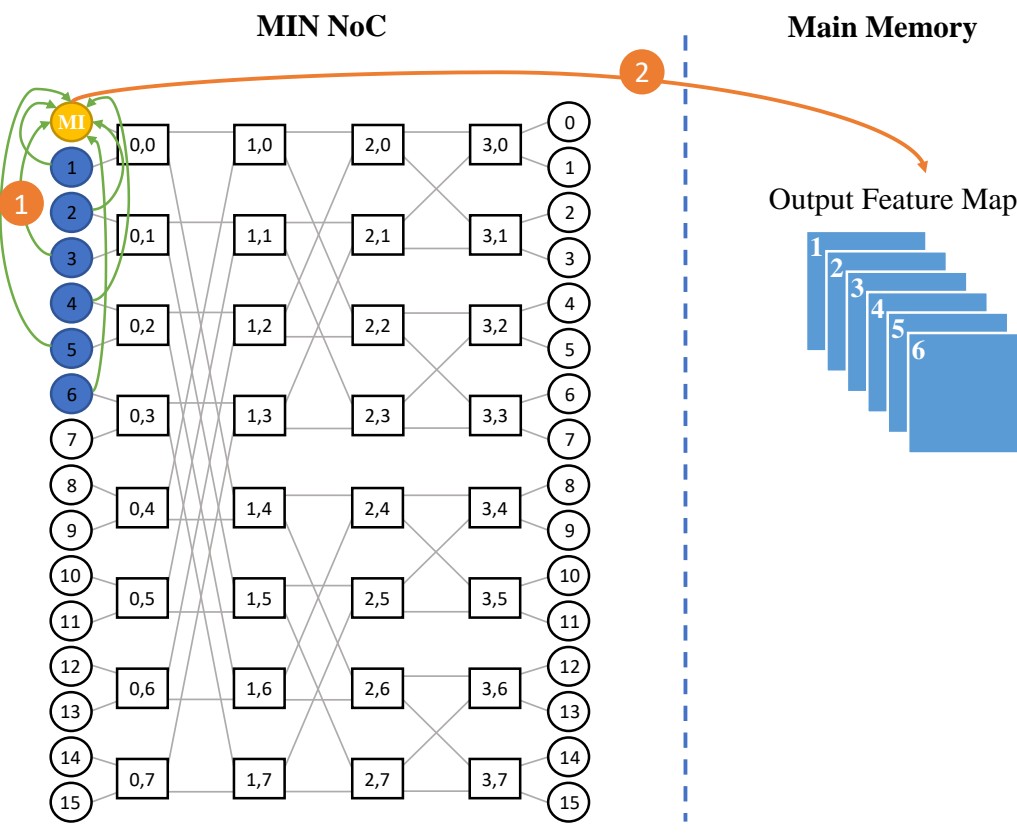

**Figure 9.** Traffic Generated in 4 × 4 Mesh for Processing Feature Maps.

The analysis is organized under the form of a function of latency and energy consumption in communication, computation and memory access (local memory and main memory access).

Figure 11 displays the average latency spent in every layer relevant to the neural network AlexNet. Regarding the memory interfaces' (MIs) locations, they are placed at the four NoC corner levels for the traditional Mesh architectures. As for the Delta MIN topology, the four MIs are uniformly spread into the MINs cores. It is worth recalling that the latency is split into three essential elements, namely, communication, computation, and memory. Clearly, for a given $NN$ and network size, the main memory access represents the predominant spot of latency for fully connected and convolutional layers concerning the Delta MIN and Mesh topologies.

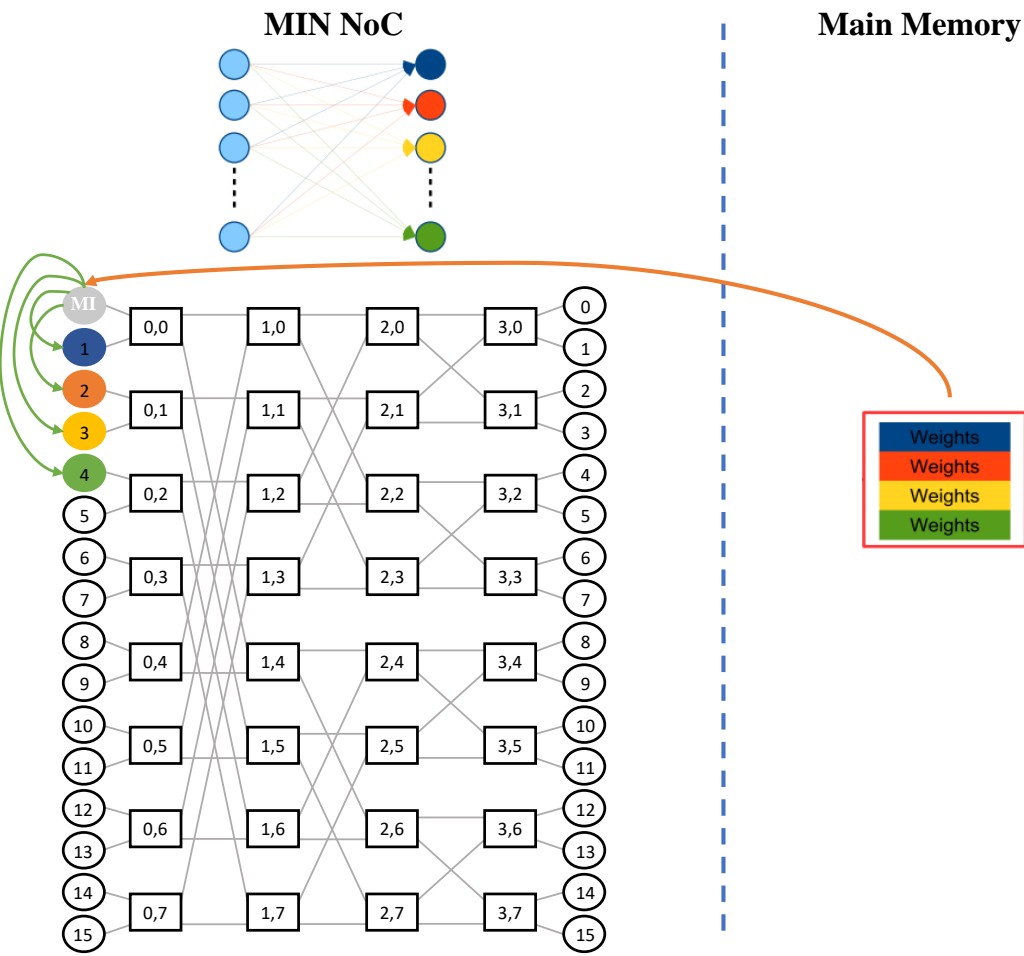

**Figure 10.** Mapping between PEs and neurons in Fully Connected Layer.

The average energy of the AlexNet DNN application mapped into two NoCs is shown in Figure 12. As the access to the main memory increases, the total energy consumption increases for the fully connected layers. The significant energy contribution of the memory is placed in the fully-connected layer as each PE requires a fetch operation from the main memory a specific amount of weights. The inversion point is meaningful for the convolutional layers. This is because, although as local memory access increases, the energy per access to the local memory slightly increases, and consequently, the global average energy decrease. Thus, there is an optimal use of the local memory element, which has an impact on the mapping of the layers on the Network-on-chip. However, the average energy values for the two NoCs show a small amount of shrinking for DELTA-based MIN.

Also, we specify two experimental platform configurations based on MIN and Mesh NoCs to vary Cores components from 64 to 1024. For the AlexNet application, the processors spread parallelism in computation in each layer, as the NoCs size increases. However, the packets routed through the network have a consequent negative impact on the communication load. As can be noted, latency tends to record slight increases with increased NoC size, along with the communication inferences and the main memory. This behavior is also valid not only for the implemented NoC model, but also the Mesh, and Delta MIN, as well as all the other Neural Network configurations (Figure 13). Noteworthy, also, is that the main memory appears to represent the predominant source of latency, followed by communication. As regards computation latency, no clear increase or decrease has been noted. It is actually this dimension feature that makes the accessibility of a large number of PEs expand the size of the NoC.

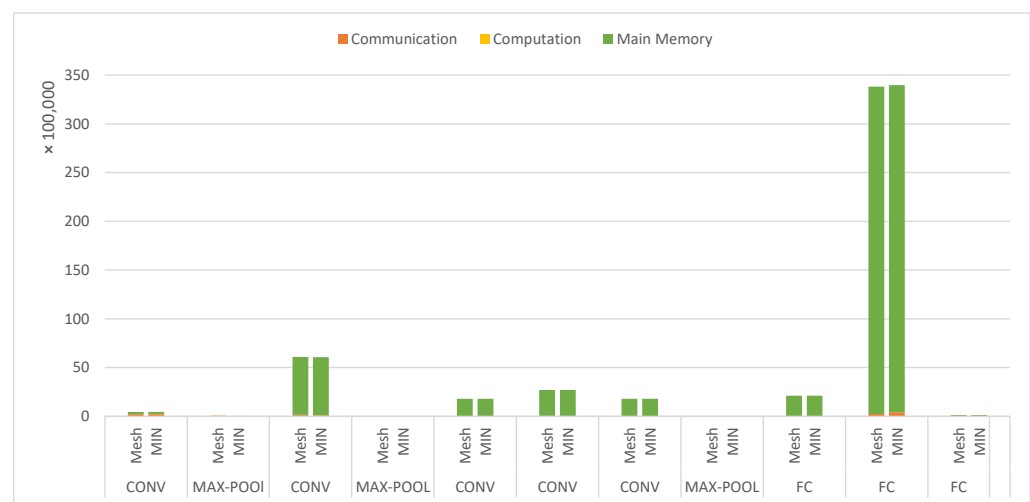

**Figure 11.** Average Latency of AlexNet over Mesh and Delta MIN topologies for 1024 NoC size.

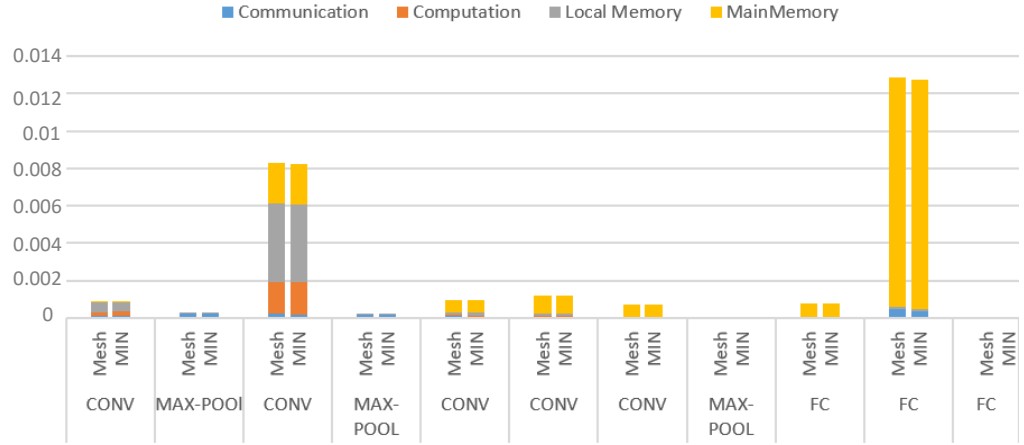

**Figure 12.** Average Energy of AlexNet over Mesh and Delta MIN topologies for 1024 NoC size.

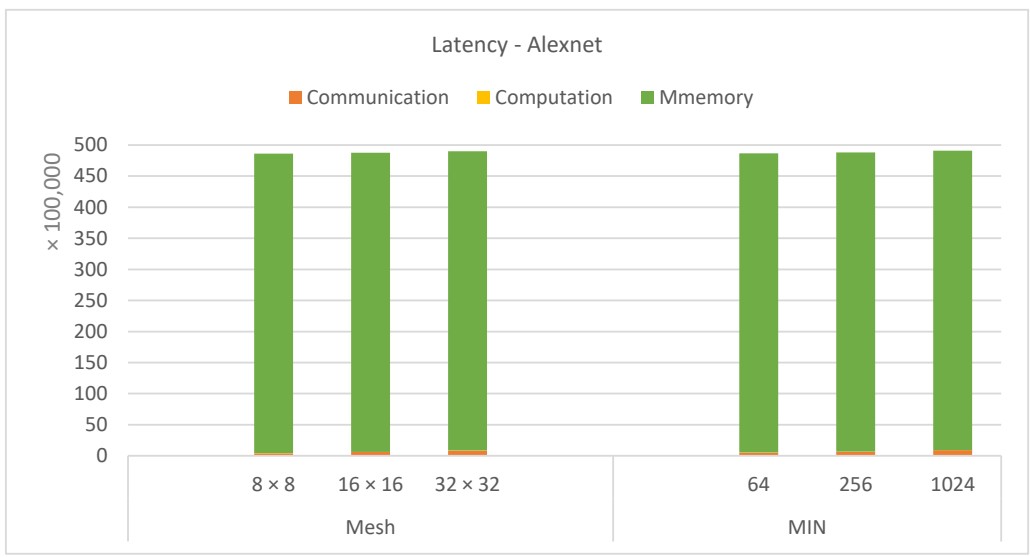

**Figure 13.** AlexNet_Total Latency for different NoC size.

## 6. Conclusions

In this research, the Design and Performance evaluation of Delta-based MINs are assessed through the process of incorporating new modules into the Noxim simulator. The regularity and scalability of these NoCs, along with their noticeable performance recorded at both of the energy and delay-associated metrics, motivated us to apply them as effective mechanisms fit for integrating Deep Neural Network accelerators. A potential research vein would involve considering larger DNNs, such as ResNet (2015), to be measured through extra architectural parameters. Additional significant features will be devoted on using these networks communication for deep learning computations, with the aim of building large-scale systems.

**Author Contributions:** Writing—original draft, Y.A.; Writing—review & editing, Y.A., S.M., F.Q.M. and M.A. All authors have read and agreed to the published version of the manuscript.

**Funding:** This research received no external funding.

**Institutional Review Board Statement:** Not applicable.

**Informed Consent Statement:** Not applicable.

**Data Availability Statement:** The data presented in this study are available on request from the corresponding author.

**Conflicts of Interest:** The authors declare no conflict of interest.

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
