# Peer review of "Mapping of Deep Neural Network Accelerators on Wireless Multistage Interconnection NoCs"

_applsci, doi:10.3390/app14010056_

Round 1

Reviewer 1 Report

1.    Parameters of network have been enhanced using training data "until the model obtains the maximum accuracy". If this accuracy is the training accuracy, maybe over-fitting has been performed. If this accuracy is the testing accuracy, the system is adjusted over the same subset that is evaluated. A validation subset could be used to optimize the system with different data than the testing data and without performing over-fitting. In addition, it would be interesting to know which range of each parameter has been analyzed."?

2.    The technical details do make much intelligibility, so please provide some strong technical details in the main methodology. The consumed time in training procedure of the proposed method and the compared algorithms can be listed.

3.    The manuscript should have a section to describe state-of-the-art techniques. This section should also outline a tabular sketch so that it is easy to identify what’s missing in the literature and how this paper addresses that. This section can be derived from the contents described in the introduction section.

4.    The results, especially the comparisons between the proposed algorithms, should be discussed more detailed. What are the insights? Why the proposed strategy/mechanism can achieve good results? All the analysis is kind of summary of the results in the tables and figures.

Extensive language is needed.

Author Response

Following your two first remarks, in this paper we are interesting to the mapping of our DNN tool into Noxim. In other words, the used DNN network, namely AlexNet, used in our investigation is already trained. The experimental platform is a simulated parameterized NoC-based Deep Neural Network that allows to assess different architectural configurations in terms of performance and energy. For the computational part, we have developed a DNN tool which has been integrated into the processing element module of Noxim cycle-accurate NoC simulator. (line 322-331)

Regarding to the state-of-the-art of mapping DNN network to Network on chip we add a paragraph describing these details from line 192 to 209 in Related work section.

For the section 5 (Evaluation Of DNN Accelerators Under Delta-Based MINs), we revised the part of experiments analysis regarding to figure 12 and 13 (line 343-364).

Reviewer 2 Report

Very good work, novel and important. The paper is organized very good and it is quite easy to follow all the concepts. However, I would suggest the following (minor) alterations: 

1. Please enlarge Figure 7 subfigures, or try to present all figures seperately (or in clusters of four subimages, because it is very difficult to understand their meaning. If it suits you, you may add a results table to summarize all your findings according to subfigures of Figure 7.

2. Please revise the quality of Figure 11, especially for the X-axis parameters.

Author Response

  • We improve the quality of figure 7 (the 3 subfigures). We can not present it in cluster of 4 images because every line in this figure presents a configuration set (32, 64 et 128 cores).
  • We revise the 3 last figures in section 5 (Evaluation Of DNN Accelerators Under Delta-Based MINs). All the X-axis are revised, we clearly distingue the MIN and Mesh topologies.

Reviewer 3 Report

Authors present an ongoing investigation, supported by their team previous work in Multistage Interconnected Networks (MIN). In this paper they propose the design and performance evaluation of a hybrid (wireless/wired) MIN and examine the implementation of our advanced DELTA-based MIN architecture on Deep Neural Network (DNN) accelerators.

Overall the paper is nicely presented and the authors explain well the subject in hand.

The main problem is that the evaluation of the DNN in section 5 appears to be lacking some further explaination and a comparison with other approaches (maybe refer them in the state-of-the-art section as well). One can not assess from the latency and energy presented by the authors if their design is appropriate or poorly performing against some other designs.

Other issues:

Reference 57 is cited in line 183 and it does not appears in the references list.

Problems with Figures:

- Figure 1, 7, 11, 12 and 13 are of poor quality

- Figure 4 mentions YMAL, do the authors refer to YAML?

- Figure 5 appears in the middle of a paragraph

Corrections needed to sentences with poor grammar or typos:

- Line 12: mainrespective 

- Lines 36-37

- Line 180

- Line 233

- Line 256

- Line 335

Author Response

  • Regarding to the state-of-the-art of mapping DNN network to Network on chip we add a paragraph describing these details from line 192 to 209 in Related work section. Also the experiments analysis has been revised (line 343-364).

  • For figures 1, 7, 11, 12 and 13 are improved in terms of quality.
  • We refer to YAML (Yet Another Markup Language).
  • Since we did some modification, the place of Figure 5 has been changed.
  • All the typos are removed
  • In regards to sentences with poor grammar are taken into consideration and improved.

Round 2

Reviewer 3 Report

I'm satisfied with the promoted changes.

I'm satisfied with the promoted changes.